# SARS-CoV-2 Serostatus and COVID-19 Illness Characteristics by Variant Time Period in Non-Hospitalized Children and Adolescents

**DOI:** 10.3390/children10050818

**Published:** 2023-04-30

**Authors:** Sarah E. Messiah, Michael D. Swartz, Rhiana A. Abbas, Yashar Talebi, Harold W. Kohl, Melissa Valerio-Shewmaker, Stacia M. DeSantis, Ashraf Yaseen, Steven H. Kelder, Jessica A. Ross, Lindsay N. Padilla, Michael O. Gonzalez, Leqing Wu, David Lakey, Jennifer A. Shuford, Stephen J. Pont, Eric Boerwinkle

**Affiliations:** 1Department of Epidemiology, Human Genetics and Environmental Sciences, School of Public Health in Dallas, The University of Texas (UT) Health Science Center at Houston, Dallas, TX 77030, USA; 2Center for Pediatric Population Health, UTHealth School of Public Health, Dallas, TX 75207, USA; 3Department of Pediatrics, McGovern Medical School, Houston, TX 77030, USA; 4Department of Biostatistics and Data Sciences, School of Public Health in Houston, The University of Texas Health Science Center at Houston, Houston, TX 77030, USA; 5School of Public Health in Austin, The University of Texas Health Science Center at Houston, Austin, TX 78701, USA; 6Department of Epidemiology, Human Genetics and Environmental Sciences, University of Texas at Austin, Austin, TX 78705, USA; 7School of Public Health in Brownville, The University of Texas Health Science Center at Houston, Brownsville, TX 78520, USA; 8Administration Division, University of Texas System, Austin, TX 78701, USA; 9Department of Medicine, The University of Texas Health Science Center Tyler, Tyler, TX 75708, USA; 10Texas Department of State Health Services, Austin, TX 78711, USA

**Keywords:** COVID-19, SARS-CoV-2, children, adolescents, symptoms

## Abstract

Objective: To describe COVID-19 illness characteristics, risk factors, and SARS-CoV-2 serostatus by variant time period in a large community-based pediatric sample. Design: Data were collected prospectively over four timepoints between October 2020 and November 2022 from a population-based cohort ages 5 to 19 years old. Setting: State of Texas, USA. Participants: Participants ages 5 to 19 years were recruited from large pediatric healthcare systems, Federally Qualified Healthcare Centers, urban and rural clinical practices, health insurance providers, and a social media campaign. Exposure: SARS-CoV-2 infection. Main Outcome(s) and Measure(s): SARS-CoV-2 antibody status was assessed by the Roche Elecsys^®^ Anti-SARS-CoV-2 Immunoassay for detection of antibodies to the SARS-CoV-2 nucleocapsid protein (Roche N-test). Self-reported antigen or PCR COVID-19 test results and symptom status were also collected. Results: Over half (57.2%) of the sample (N = 3911) was antibody positive. Symptomatic infection increased over time from 47.09% during the pre-Delta variant time period, to 76.95% during Delta, to 84.73% during Omicron, and to 94.79% during the Omicron BA.2. Those who were not vaccinated were more likely (OR 1.71, 95% CI 1.47, 2.00) to be infected versus those fully vaccinated. Conclusions: Results show an increase in symptomatic COVID-19 infection among non-hospitalized children with each progressive variant over the past two years. Findings here support the public health guidance that eligible children should remain up to date with COVID-19 vaccinations.

## 1. Introduction

As of 6 April 2023, almost 15.6 million children in the United States have tested positive for COVID-19 since the onset of the pandemic [1]. The number of new pediatric COVID-19 cases remained high through Spring 2022, suggesting higher transmissibility of the Omicron B1.1.529 variant versus the Delta B.1.617.2 and Alpha B.1.1.7 variants, among others [2]; however, the true incidence of SARS-CoV-2 infection in children and adolescents not hospitalized with COVID-19 illness is not known due to the high proportion of asymptomatic infection and the prioritization of testing for older adults and those with severe illness that occurred early in the pandemic [3,4]. Additionally, pediatric COVID-19 hospitalization rates are significantly lower than those for adults, further suggesting that children have less severe illness from COVID-19 compared to adults [1]; however, among hospitalized children, one-third are admitted to intensive care where approximately six percent receive mechanical ventilation [5], and a very small number develop Multisystem Inflammatory Syndrome in Children (MIS-C), which includes hypotension, severe abdominal pain, and cardiac dysfunction [6,7].

During the first year of the pandemic, approximately 20% of COVID-19 adult cases were asymptomatic [8], as well as an estimated 30–50% of pediatric cases [9]. A review of mostly hospitalized pediatric patients earlier in the pandemic found fever and cough were the most common COVID-19 symptoms, with other symptoms being infrequent [10,11]. One study in the UK reported the seven most prevalent symptoms were common to both Alpha and Delta variants [12]; however, little is known about the presence of symptoms among non-hospitalized pediatric populations with SARS-CoV-2 antibodies through all three dominant (pre-Delta, Delta, Omicron) waves. The World Health Organization’s (WHO) population-based age-stratified sero-epidemiological investigation protocol for COVID-19 infection [13] has two primary objectives: (1) to measure the seroprevalence of antibodies to COVID-19 in the general population to ascertain the cumulative immunity; and (2) to estimate the proportion of asymptomatic, pre-symptomatic, or subclinical infections in the population, and by key demographics. Following this guidance, the purpose of this investigation was to estimate the prevalence of SARS-CoV-2 antibody status and characterize symptoms in a cohort of children and adolescents ages 5 to 19 years who were not hospitalized with COVID-19 illness, stratified by periods that were dominated by Alpha and other variants (pre-Delta) versus the Delta and Omicron variants. A secondary aim was to define risk factors for symptomatic infection. It was hypothesized that >30% of those with SARS-CoV-2 antibody positive status would be asymptomatic for all variants.

## 2. Materials and Methods

### 2.1. Study Design

Texas CARES (Coronavirus Antibody REsponse Survey) is an ongoing prospective population-based survey of a volunteer sample from the general population that includes participants ages 5 to 90 years. Participants can receive up to four free SARS-CoV-2 antibody tests, two to three months apart. The current study utilizes a repeated measures sample of the 4 antibody tests among the rolling recruitment of participants over a two-year period (October 2020 to October 2022). Texas CARES is a partnership between the University of Texas Health Science Center at Houston School of Public Health, Texas Department of State Health Services (DSHS), the University of Texas System, and Clinical Pathology Laboratories (CPL), a laboratory facility with more than 200 statewide sites. All protocols were reviewed and approved by the University of Texas Health Science Center’s Committee for the Protection of Human Subjects, but also deemed public health practice by the Texas Department of State Health Services IRB.

### 2.2. Study Population

Texas CARES began enrolling participants across the state of Texas in October 2020. Families of potential pediatric participants were informed about the survey in several ways: via their healthcare provider (included providers in large pediatric healthcare systems, Federally Qualified Healthcare Centers, and urban and rural clinical practices), insurance carrier if they were Medicaid insured, media (radio, billboard) spots, social media campaigns, community events, and word of mouth. All information was delivered in English and Spanish.

### 2.3. Study Procedures

A parent or designated caregiver served as the proxy informed consent for children and adolescents to participate in Texas CARES. Adolescents > 12 years old had the option to sign assent and complete the questionnaire. No adolescents refused to provide assent or participate. Participants who consented to enroll in Texas CARES first completed a short online questionnaire designed to collect demographic information, employment, baseline medical conditions and comorbidities, prior COVID-19 tests and diagnoses, physician diagnosis of COVID-19 and other chronic illnesses (e.g., type 2 diabetes, asthma, hypertension), and previous COVID-19 symptoms and severity. Once the participant completed the survey, orders were generated to bringto a laboratory facility of their choice to complete the antibody status blood draw. Participants typically received their results within 48 h.

### 2.4. Study Measures

SARS Cov-2 Antibody Assay Roche Diagnostics. Antibody status was assessed via the Roche Elecsys^®^ Anti-SARS-CoV-2 Immunoassay. The Roche assay detects high-affinity antibodies to SARS-CoV-2 using a modified recombinant protein representing the nucleocapsid (N) antigen for the determination of SARS-CoV-2 antibodies [14]. The assay relies on a double antigen sandwich (DAGS) format that enriches detection of higher affinity antibodies, which are more likely to be specific for SARS-CoV-2. The assay format is agnostic to the antibody isotype and can detect high affinity antibodies of all isotypes; it preferentially detects IgG antibodies since these are most likely to evolve to become high affinity but can also detect IgM and IgA antibodies. The nucleocapsid antigen is abundantly expressed and is a useful target for sensitive detection of virus-specific antibodies. These features provide an optimal combination of high specificity and sensitivity for the detection of immune exposure to SARS-CoV-2 in the general population, including pregnant women and pediatric populations. The test has a published sensitivity of 99.5% (95% CI: 97.0–100) and 99.8% specificity (95% CI: 99.69–99.88) in diagnostic specimens (n = 2861) [14,15].

*Electronic Questionnaire.* An online, REDCap [16,17]-programmed questionnaire was designed to be completed in 10–15 min to capture previously described information by the parent proxy. To increase validity and reproducibility, most questions and response formats were replicated from the COVID-19 PhenX Toolkit [18] and Behavioral Risk Factor Surveillance System questionnaires [19]. US Census race/ethnicity questions were also replicated [20]. The self-administered questionnaire was designed to be completed after signed informed consent using a seamless webpage transition to ease respondent burden and maximize survey completion.

*Body Mass Index*. Body weight categories were determined using the calculated body mass index (BMI; kg/m^2^) from caregiver-reported height and weight of the child, which was then transformed to a standardized percentile distribution based on the Centers for Disease Control and Prevention (CDC) age- and sex-adjusted BMI growth charts [21]. Standardized weight categories are as follows; (1) underweight ≤ 5th percentile; (2) healthy weight = 5th–<85th percentile; (3) overweight ≥85th–<95th percentile; and (4) obesity ≥ 95th percentile [22].

*Definition of Variant Time Periods and Infection Status*. The pre-Delta-variant(s) time period was defined as any infection before 5 January 2021 (inclusive), the Delta-variant time period was from 1 July 2021 to 30 November 2021, the Omicron-variant time period was from 25 December 2021 to 19 March 2022, and the Omicron-BA.2-variant time period was from 26 March 2022 onward. Times between variants (1 May 2021–30 June 2021, 1 December 2021–24 December 2021, and 109 March 2022–25 March 2022) were considered to be variant cross-over periods and thus all results during these weeks were excluded from this analysis. Variant time periods were verified by Texas Department of State Health Services genetic sequencing data. Infection status was self-reported as confirmed by PCR test or doctor diagnosis and then confirmed with a positive antibody test. A total of 155 self-reported SARS-CoV-2 infections were removed from the analysis because they could not be confirmed with a positive antibody status, and thus, we could not attribute symptoms specifically to COVID-19 illness (versus influenza, respiratory syncytial virus [RSV], for example).

*Definition of Symptomatic Status*. Participants self-reported symptom status (Y/N) from the following list: difficulty breathing, feeling tired all the time, brain fog or cognitive impairment, cough, chest pain, headache, racing heart rate, joint pain, muscle pain, tingling or burning pain in fingers and toes, abdominal pain, diarrhea, insomnia or other sleep difficulties, fever, lightheadedness, impaired daily function and mobility, rash, mood changes, new loss of taste and/or smell, and menstrual cycle irregularities.

### 2.5. Statistical Analysis

Standard descriptive statistics (age, sex, race, ethnicity, residential density area, body mass index group, serostatus, number of SARS-CoV-2 infections) summarized pediatric participant characteristics. Continuous variables were presented as means and standard deviations, and categorical variables were calculated as frequencies and percentages. Repeated measures analysis where each survey counted as an observation with multiple observations (up to 4) within individuals was the primary statistical approach and design. Logistic regression with a generalized estimating equation (GEE) framework was used to account for within individual correlations among repeated observations. First, infection status (+/−) was modeled as the outcome. When an individual reported in their survey response that they had a new infection since the previous survey, and with a date of infection, the infection outcome was considered positive, and if they did not report any new infection, their outcome was coded as negative at the date of the survey. Dates (of each outcome) were used to classify the outcome into a variant time period. As such, each participant could have up to 4 outcomes.

In the Generalized Estimating Equations (GEE) framework, quasi-likelihood information criteria (QIC)23 was used to select the correlation matrix structure and best model fit. The correlation matrix selected was independent. BMI, urban/rural residential status, age, ethnicity, and sex were not selected for but were fixed due to research interest. Computations were implemented in R using the QIC.geeglm command.

To model symptomatic status, the analysis was first limited to those who reported at least one infection, and each infection was classified within individuals as symptomatic or asymptomatic based on self-reported symptom variables. The same GEE modeling strategy was completed for repeated measures as outlined above, using symptomatic status as the outcome.

The same variable selection strategy was applied and included the same covariates as above, with the addition of vaccination status for the symptomatic model. Computations were implemented in R using the “geepack” package.

## 3. Results

The final analytical sample consisted of 3911 children and adolescents (50.6% female, mean age 12.8 years [SD 3.8], 22.1% Hispanic, 2.4% non-Hispanic Black). Most reported that they lived in an urban area (90.2%). The majority of participants were at a healthy weight (59.5%), 15.0% had overweight and 14.5% had obesity. About one-third reported having a previous COVID-19 positive oral/nasal test or a positive antibody test (33.8%). The majority of the sample (57.2%) had a positive SARS-CoV-2 antibody N antibody test. Additionally, the majority (66.2%) of the sample reported no previous SARS-CoV-2 infection, while 29.7% reported one infection and 3.7% reported two infections. Slightly more than a third (37.74%) of the sample reported being fully vaccinated, while 59.52% reported being partially vaccinated, and 2.74% reported no vaccination. Of those who reported COVID-19 disease symptoms, 18.67% stated they were mild and less than 1% reported severe symptoms (Table 1). Appendix A shows vaccination status by each of the four variant time periods, and Appendix A shows COVID-19 disease symptom severity by variant time period.

A total of 26.40% (429/1196) reported an infection (symptomatic or asymptomatic) during the pre-Delta-variant time period, 9.94% (322/2916) during the Delta-time period, 18.95% (297/1270) during the Omicron-time period, and 7.72% (100/1195) during the Omicron-BA.2-variant time period. 

Less than half (47.09%) of child and adolescent participants with positive serostatus reported being symptomatic during the pre-Delta-variant time period, and this increased to 76.95% during Delta, 84.73% during Omicron, and 94.79% during Omicron BA.2. Of those who were diagnosed with COVID-19 by antigen or PCR test during the pre-Delta-time period, the five most common symptoms or symptom combinations reported were (1) new loss of taste and/or smell; (2) headache; (3) a combination of headache, fatigue, congestion, fever, cough, sore throat, and aches; (4) fever; and (5) congestion. (Figure 1a). During the Delta-variant timeframe, the five most reported symptoms were (1) a combination of congestion, fatigue, headache, fever, cough, sore throat, and aches; (2) congestion; (3) the previous reported combination of symptoms plus new loss of taste and/or smell; (4) congestion and cough; and (5) congestion, cough and sore throat. (Figure 1b). During the Omicron-variant timeframe, the five most reported symptoms were (1) a combination of congestion, headache, fatigue, cough, sore throat, fever, and aches; (2) congestion; (3) congestion and headache; (4) congestion and sore throat; and (5) congestion, headache, fatigue, sore throat, cough, and aches. (Figure 1c). The most common symptoms during the Omicron-BA.2-variant time period included a combination of cough, fever, sore throat, fatigue, headache, congestion, and aches. (Figure 1d).

Logistic regression results showed that participants were over two and a half times more likely to be infected during the pre-Delta-(Odds Ratio [OR] 2.65, 95% CI, 2.25, 3.12) and Omicron (OR 2.50, 95% CI, 2.11, 2.96)-variant time periods versus the Delta-variant time period. Males were less likely to report being infected (OR 0.87, 95% CI, 0.77, 0.99) versus females. Older children ages 10 to 14 years (OR 1.34, 95% CI 1.12, 1.59) and ages 15 to 19 years (OR 1.41, 95% CI 1.17, 1.69) were more likely than 5-to-9-year-olds to be infected with SARS-CoV-2. Those who were non-Hispanic (OR 1.22, 95% CI, 1.04, 1.43) or not vaccinated (OR 1.71, 95% CI 1.47, 2.00) were more likely to be infected versus those who were Hispanic and those fully vaccinated, respectively. Those who reported one previous infection were less likely to report being infected (OR 0.68, 95% CI, 0.54, 0.85) versus those who reported no previous infection. (Table 2).

Logistic regression results showed that those infected during the Omicron-variant time period (OR 1.73, 95% CI 1.04 2.90) and Omicron-BA.2-variant time period (OR 5.81, 95% CI, 2.19, 15.37) were more likely to be symptomatic while those infected during the pre-Delta-variant time period were less likely to be symptomatic (OR 0.25, 95% CI 0.18, 0.34) versus those infected during the Delta-variant time period (Table 3).

## 4. Discussion

We report one of the world’s largest and most current pediatric COVID-19 sero-epidemiological samples, and one of the first representing the pre-Delta-, Delta- and Omicron-variant time periods over the past two years. Results show that over time the proportion of symptomatic infection increased as variants mutated. Specifically, there was a doubling of symptomatic infection from 47% pre-Delta to 95% during the Omicron-BA.2-variant time period. Symptoms changed over time as well with a new loss of taste and/or smell only being prevalent early in the pandemic. Significant risk factors for infection included being infected with pre-Delta or Omicron variants, being of older age, non-Hispanic, and not being fully vaccinated. These results are an important benchmark to understanding how multiple COVID-19 waves and SARS-CoV-2 variants have affected the US pediatric population over time and through the first two years of the pandemic.

To give Texas-specific public health mitigation strategy context to the findings reported here, on 19 March 2020, Executive Order No. GA-08 was issued by Governor Gregg Abbot, effective March 21 through April 3 with restrictions on certain social activities. Additionally, on 19 March 2020, the Texas Department of State Health Services determined that COVID-19 represents a public health disaster pursuant to the Texas Health and Safety Code. By 1 April 2020 more than 50 Texas counties had issued stay-at-home orders, with many clusters for counties around the most populated Texas cities. This was followed by Gov. Abbott clarifying his previously issued executive order, saying that it “requires all Texans to stay at home” except for essential activities starting on 2 April 2020. Approximately 1 month later on 1 May 2020, the first phase of the Governor’s plan to allow retail stores, restaurants, malls, and movie theaters to reopen at 25% capacity began, followed by subsequent phases to completely reopen the state (a detailed timeline can be found at https://www.huschblackwell.com/texas-state-by-state-COVID-19-guidance, accessed at 23 February 2023). Schools remained closed for the school year but reopened in August for the 2020–2021 school year with a virtual option available to families. The following two school years saw a subsequent complete relaxation of social distancing and masking requirements.

The above fairly progressive timeline of public health mitigation strategy reversals may be why our seroprevalence results for the pre-Delta-variant time period do not generally align with the other literature [23,24,25,26,27,28,29,30,31,32,33] but are similar to one study in Israel that reported 34% of children with positive status [34]. This is most likely due to the majority of the published reports collecting data early in the pandemic or in 2020 only. Texas opened schools for both the 2020–2021 and 2021–2022 school years, as well as many sports programs and other activities, and results here suggest the virus was widely circulating in these settings during these time periods and especially during the time when Omicron became the dominant variant. One US study of summer camps showed a thirty-one-fold increase in COVID-19 cases compared with June–July 2020 in Louisiana, evidence of the Delta variant’s impactful contagion compared to pre-Delta [33] and is in disagreement with our findings here. The only large scale pediatric-only estimates in the US to date reported the prevalence of ~1% among 1076 residual serum samples of children seeking medical care in Seattle [24], 16.3% among 1603 residual blood samples of persons < 18 years old in Mississippi [25], and 1.71% in 555 residual blood samples in Missouri [26]. All of these samples were tested in March–September of 2020, when the epidemic was in the first and second wave in the US, thus perhaps explaining the lower prevalence estimates. Internationally, in July/August 2020 in a cohort of 200 children who attended the pediatric department of a large hospital in Prague, Czech Republic, not a single case of seropositivity was found [27]. Similarly, a study in the first wave of the pandemic in Germany of 2482 children (median age, 6 [range, 1–10] found strikingly low seroprevalence in children (0.4%). Seroprevalence was also low in 2482 parents (median age, 40 [range 1.8%, 95% CI, 1.2–2.4%]) [28].

Although there are more than 100 published SARS-CoV-2 seroprevalence studies, either very few include children, they represent a small proportion of the overall sample [29,30,31,32,33], or represent hospitalized samples only [34,35,36,37,38]. One exception is a household sample survey completed in Spain in the first wave of the pandemic that included 6527 completed immunoassay results in 0-to-19-year-olds (3.2% seropositivity, 95% CI, 3.2–4.6) [39]. In general, studies have shown younger children in particular have a lower risk of infection versus adults. For example, in a population-based study in Switzerland, 5-to-9-year-olds had a significantly lower risk of being seropositive than those aged 20 to 49 years (RR 0.32, 95% CI 0.11–0.63) [31]. Another population-based sample in southwest Germany early in the pandemic (Spring 2020) that included children aged 1 to 10 years and a corresponding parent reported the estimated SARS-CoV-2 seroprevalence was low in parents (1.8%) and three-fold lower in children (0.6%) [28]. Finally, another study in England that collected seroprevalence data from October 2019 through June 2021 reported that 32.7% of participants aged 15 to 18 years old had evidence of antibodies [40]. These results are quite similar to our results here that found that older ages had significant increased risk of infection versus younger children.

A second area of key novel findings reported here showed that over four SARS-CoV-2-variant time periods; the proportion of symptomatic infection increased as variants mutated. To the authors’ knowledge, this has not been previously reported in the literature among children not hospitalized with COVID-19 illness. One recently published study using medical claims data in Israel during the third and fourth waves of the pandemic (1 December 2020 to 30 April 2021, and 1 June 2021 to 10 October 2021, respectively) reported that the rate of children with symptomatic disease among patients with documented SARS-CoV-2 infection was higher in the fourth wave compared to the third wave (49.9% vs. 37.5%) [41]. They also reported, along with other early pandemic studies [42,43] that fever as the most common and increasing symptom (33% during the fourth wave vs. 13.6% in the third wave), whereas findings here showed fever was more likely to be reported as a component of a constellation of simultaneous symptoms. Specifically, results here showed the most common pre-Delta symptoms or symptom combinations reported were (1) new loss of taste and/or smell; (2) headache; (3) a combination of headache, fatigue, congestion, fever, cough, sore throat, and aches; (4) fever; and (5) congestion. This changed during the Delta-, Omicron-, and Omicron-BA.2-variant time periods to a combination of congestion, fatigue, headache, fever, cough, sore throat, and aches becoming the most prevalent symptoms reported. Others have reported similar findings in terms of the constellation of symptoms but earlier in the pandemic. Specifically, one study (December 2020–July 2021) of 109,626 school-aged children in the United Kingdom showed the seven most prevalent symptoms were common to both the Alpha or pre-Delta and Delta variants [44]. The odds of presenting several symptoms were higher with Delta than Alpha infection, including headache and fever. Additionally, of note, the Israeli study mentioned above reported that preschool-aged children had the lowest prevalence of illness compared to other age groups. While our study did not include this age group, our findings were similar in that 10-to-14-year-olds and 15-to-19-year-olds were significantly more likely than 5-to-9-year-olds to be infected with SARS-CoV-2.

Our results showed that pediatric participants who are not fully vaccinated are at an increased risk for infection. This finding aligns with other worldwide studies among adults [45,46] and children ages 5–11 years [46], as well as non-immunocompromised children [47]. As such, it has been recommended that all eligible children should remain up to date with COVID-19 vaccinations, especially as influenza and RSV are also widely circulating in the general population [47].

In summary, findings reported here contribute to the current knowledge on pediatric SARS-CoV-2 seroprevalence is several novel ways. First, this analysis is one of the only pediatric longitudinal cohorts in the literature and in the United States that has included data capture across four variant time periods in the pandemic, and that started in 2020 and is still ongoing. While there are other seroprevalence studies published as previously described, few also capture symptom and vaccine status over time and most previous studies occurred early in the pandemic or before the Omicron variant became predominant. This information is important for both clinical management and public health measures, especially as influenza and respiratory syncytial virus (RSV) as well as other common childhood viruses are simultaneously circulating in the population. In other words, it is important for pediatricians and other healthcare providers to have documentation of what symptoms a SARS-CoV-2 infection in a child may be causing now versus earlier in the pandemic to potentially rule out (or in) infection, and how these symptoms may mimic those common to influenza, RSV, or other viral infections. This is particularly important information for healthcare providers who have pediatric patients who have never been hospitalized due to COVID-19 illness and, thus. may not be aware of how SARS-CoV-2 infection symptoms have substantially changed from the beginning of the pandemic. Second, Texas CARES has uniquely captured vaccination uptake data over time that is linked to demographic information. These findings can inform public health vaccination campaigns in terms of populations at high risk for future infection. Our data showed a substantial proportion of children are not protected against future SARS-CoV-2 infection by either naturally acquired antibodies or vaccine-induced, so public health efforts should continue to promote vaccination as a preventive measure against future COVID-19 illness. Third, findings reported here highlight the importance of public health surveillance as an important tool to identify how SARS-CoV-2 seropositivity changes over time in the general population. As we have shown here, the pandemic has evolved in many ways in terms of variants, symptoms produced, and the introduction and uptake of vaccines, and these findings are essential to inform future policy recommendations for not only the current pandemic but others we may be facing in the future.

### Study Limitations

Findings presented here should note specific limitations. Because the study design was based on non-random sampling of participants, unweighted analysis cannot provide precise estimates of seroprevalence by age group in the general Texas population. Additionally, of note, while the study sample represents a large, demographically diverse, and geographically spread state, participants self-selected to participate, which may result in selection biases that limit generalizability to the entire population. A second limitation due to the sampling frame is another form of selection bias; that is, parents who suspected their child or an adult in the household was infected with SARS-CoV-2 may have been more likely to participate; however, a substantial proportion of children reported being asymptomatic across all variants. A third limitation is that while the Roche Elecsys^®^ Anti-SARS-CoV-2 Immunoassay does detect the prevalence of IgM, IgA, or IgG antibodies, it is not able to parse out the specific immunoglobulin antibody status individually. A fourth limitation of the current analysis is the exclusion of participants less than 5 years old, and thus, any result reported here cannot be generalized to that age group. Finally, genomic sequencing was not performed to genetically characterize pre-Delta and other multiple Omicron variants.

## 5. Conclusions

Self-reported data among those who received both antibody and antigen or PCR tests in each variant timeframe suggests that SARS-CoV-2 has been widely circulating in the pediatric population and becoming more symptomatic over time. Not being vaccinated, older age, and non-Hispanic are risk factors for infection. It will be important to continue to monitor symptoms over time, especially as influenza and RSV have also been widely circulating in the US pediatric population.

## Figures and Tables

**Figure 1 children-10-00818-f001:**
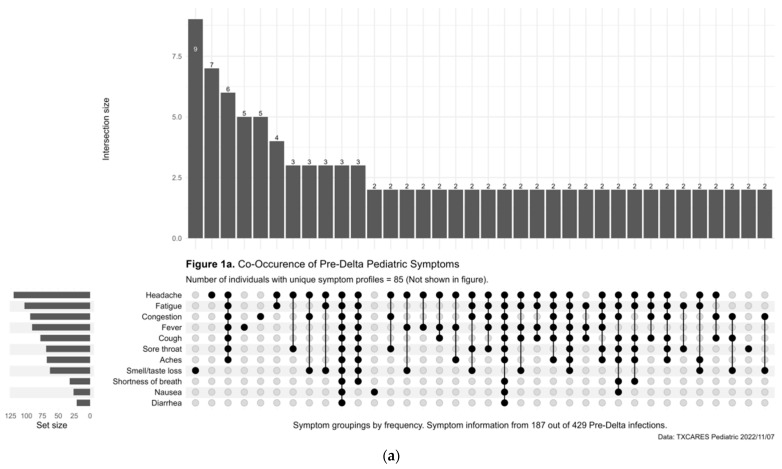
(**a**) Upset Plot, pre-Delta-variant time period, symptom frequency. (**b**) Upset Plot, Delta-variant time period, symptom frequency. (**c**) Upset Plot, Omicron-variant time period, symptom frequency. (**d**) Upset Plot, Omicron-BA.2-variant time period, symptom frequency.

**Table 1 children-10-00818-t001:** Descriptive and COVID-19-Related Characteristics of the Texas CARES Pediatric Sample (N = 3911).

Age	
Mean (SD)	12.8 (3.8)
Age Group, Years, n (%)	
5-to-9 year olds	863 (22.1)
10-to-14 year olds	1588 (40.6)
15-to-19 year olds	1460 (37.3)
Sex, n (%) ^1^	
Female	1980 (50.6)
Male	1926 (49.2)
None of these describe me	1 (0.0)
Ethnicity, n (%) ^2^	
Hispanic	865 (22.1)
Non-Hispanic	2949 (75.4)
Unknown/Other	57 (1.5)
Race, n (%) ^3^	
Non-Hispanic White	3275 (83.7)
Non-Hispanic Black	92 (2.4)
Asian	222 (5.7)
Multiracial	196 (5.0)
American Indian or Alaskan Native	18 (0.5)
Hawaiian or Other Pacific Islander	4 (0.1)
Residential Density, n (%)	
Urban	3526 (90.2)
Rural	385 (9.8)
Body Mass Index, n (%) ^4^	
Underweight	162 (4.1)
Healthy	2326 (59.5)
Overweight	587 (15.0)
Obesity	569 (14.5%)
N Antibody Status	
Negative	1673 (42.8)
Positive	2238 (57.2)
Number of Reported Infections, n (%)	
0	2591 (66.2)
1	1161 (29.7)
2	145 (3.7)
≥ 3	14 (0.3)
Vaccination Status	
Full	1476 (37.74)
Partial	2328 (59.52)
None	107 (2.74)
COVID-19 Disease Severity	
Mild	730 (18.67)
Severe	14 (0.36)
Missing	3167 (80.98)

^1^ Data were missing for 4 children, ^2^ Data were missing for 40 children, ^3^ Data were missing for 104 children, ^4^ Data were missing for 267 children.

**Table 2 children-10-00818-t002:** Odds of SARS-CoV-2 Infection by Descriptive Characteristics (n = 3906).

	Odds Ratio (95% CI) ^a^	*p* Value
Variant period		
Delta	REF	–
Pre-Delta	2.65 (2.25–3.12)	<0.001
Omicron	2.50 (2.11–2.96)	<0.001
Omicron BA.2	0.95 (0.75–1.21)	0.688
Age groups, years		
Ages 5-to-9	REF	–
Ages 10-to-14	1.34 (1.12–1.59)	0.001
Ages 15-to-19	1.41 (1.17–1.69)	<0.001
Sex ^b^		
Female	REF	–
Male	0.87 (0.77–0.99)	0.037
Ethnicity		
Hispanic	REF	–
Non-Hispanic	1.22 (1.04–1.43)	0.014
Missing	1.27 (0.81–2.00)	0.294
BMI, categorical ^c^		
Healthy	REF	–
Underweight	1.02 (0.74–1.39)	0.925
Overweight	1.01 (0.85–1.21)	0.879
Obesity	1.16 (0.97–1.39)	0.110
Missing	0.95 (0.72–1.26)	0.731
Residential density		
Urban	REF	–
Rural	0.99 (0.80–1.24)	0.956
Vaccination status		
Fully vaccinated	REF	–
Unvaccinated	1.71 (1.47–2.00)	<0.001
Partially vaccinated	1.05 (0.68–1.62)	0.830
Number of previous infections		
None	REF	–
One	0.68 (0.54–0.85)	<0.001
Two or more	0.68 (0.38–1.24)	0.209

^a^ Multavariable logistic regression adjusted for SARS-CoV-2 variant, age group, sex, ethnicity, BMI group, residential status, vaccination status, and number of previous infections. ^b^ Data were missing (n = 4) or participant did not identify as either male or female (n = 1), thus overall sample size reduced by (n = 5).^c^ Body Mass Index (BMI), healthy ≤ 85th %ile; overweight ≥ 85th %ile – < 95th %ile; obesity ≥ 95th %ile, all adjusted for age and sex [22].

**Table 3 children-10-00818-t003:** Odds of Symptomatic Infection by SARS-CoV-2 Variant and Descriptive Characteristics (n = 1189).

	Odds Ratio (95% CI) ^a^	*p* Value
Variant period		
Delta	REF	–
Pre-Delta	0.27 (0.19–0.38)	<0.001
Omicron	1.73 (1.04–2.90)	0.036
Omicron BA.2	5.81 (2.19–15.37)	<0.001
Sex		
Female	REF	–
Male	0.77 (0.58–1.03)	0.073
Ethnicity		
Hispanic	REF	–
Non-Hispanic	1.15 (0.81–1.64)	0.697
Missing	1.25 (0.45–3.47)	0.674
BMI, categorical ^b^		
Healthy	REF	–
Underweight	1.20 (0.56–2.54)	0.541
Overweight	1.17 (0.77–1.75)	0.460
Obesity	1.21 (0.79–1.84)	0.379
Missing	0.59 (0.31–1.11)	0.103
Residential density		
Urban	REF	–
Rural	0.79 (0.50–1.25)	0.317
Vaccination status		
Fully vaccinated	REF	–
Unvaccinated	1.11 (0.65–1.90)	0.881
Partially vaccinated	1.19 (0.38–3.69)	0.768

^a^ Multavariable logistic regression adjusted for SARS-CoV-2 variant, sex, ethnicity, BMI group, residential status, and vaccination status. ^b^ Body Mass Index (BMI), healthy ≤ 85th %ile; overweight ≥ 85^th^%ile – < 95th %ile; obesity ≥ 95th %ile, all adjusted for age and sex [22].

## Data Availability

Texas CARES investigators are committed to data sharing. Granular results and user-specified data summaries are currently publicly available on the Texas CARES portal (https://sph.uth.edu/projects/texascares/dashboard, accessed on 20 February 2023). When baseline recruitment is complete, a deidentified individual level dataset will be available for download from the same portal.

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
