# Peer review of "SARS-CoV-2 Serostatus and COVID-19 Illness Characteristics by Variant Time Period in Non-Hospitalized Children and Adolescents"

_children, 2023, doi:10.3390/children10050818_

Round 1

Reviewer 1 Report

Sarah E. Messiah and colleagues reported the SARS-Cov-2 serostatus study in children and adolescents in Texas. The data was collected from 4 different time periods: pre-Delta, Delta, Omicron and Omicron BA.2. The results showed that SARS-Cov-2 had been widely circulating in the pediatric population and becoming more symptomatic over time. The study is well-designed and the experimental approaches and the interpretation of the data are appropriate. I only have a few questions here.

1: Do you have the data on the children being vaccinated or not? If they were vaccinated, the authors need to discuss it.

2: What does the REF mean in table 2?

3: what is the X-axis for figure 1?

4: Please indicate where the data from in discussion lines 34-35.

Author Response

Reviewer 1

Comment: Do you have the data on the children being vaccinated or not? If they were vaccinated, the authors need to discuss it.

Response: Based on this comment we have added vaccination status to Table 1 and in the summary is provided in the first paragraph of the results. We have also added a new Supplemental Table 1 that summarizes vaccination status across the four variant time periods.

Comment: What does the REF mean in table 2?

Response: REF denotes reference category which is standard for reporting of logistic regression analyses. In other words, this is the group that all others were compared to in the reported models.

Comment: What is the X-axis for figure 1?

Response: The X axis for Figure 1 is the specific COVID-19 illness symptoms reported, either individually or in combination as designated immediately below the title of the axis.

Comment: Please indicate where the data from in discussion lines 34-35.

Response: Lines 34-35, stated in the abstract as “Children were over twice as likely to have a positive antibody test during the pre-Delta (Odds Ratio [OR] 2.65, 95% CI, 2.25, 3.12) and Omicron (OR 2.50, 95% CI, 2.11, 2.96) variant time periods versus Delta, respectively” are from Table 2, the first few rows of data reported summarizing the odds of SARS-CoV-2 infection by descriptive characteristics, and in this case by variant time period.

Reviewer 2 Report

Thank you for asking me to review the manuscript entitled “SARS-CoV-2 Serostatus and COVID-19 Illness Characteristics 2 by Variant Time Period in Non-Hospitalized Children and Adolescents” written by Messiah et al. The manuscript describes differences in clinical characteristics in non-hospitalised youth diagnosed with SARS-CoV-2 over four timepoints with the main outcome measured being serostatus. The manuscript is well written although the conclusions are not surprising given the epidemiologic history of SARS-CoV-2 and similar findings have already been reported, although less data on non-hospitalised patients is available. The more interesting finding is that those not vaccinated were more likely to be seropositive, albeit also not a surprising or novel. This does add to the evidence supportive of vaccination against SARS-CoV-2 in the paediatric population.

Minor comments:

-population data for those <5 yo would be important particularly given increased prevalence of infection in young children in Omicron waves and important epidemiologically over time. Is this simply a limitation of the data set?

-how were the time periods of variant determined. Were they local rates of dominant variant >50% in Texas, in the USA, worldwide?

-is a doctor diagnosis reliable especially in subsequent waves with increased community circulation of other viruses (and with such broad definition of symptomatic and less public health measures in Texas emphasising social distancing with open schools and extracurricular activities)? Confirmation with a positive antibody test in acute infection may still be in the window period before antibody development (this seems plausible for those who were actually excluded), if positive antibody this could be past infection with current symptoms representative of a non-SARS-CoV-2 related illness. Were these also confirmed by antigen or PCR testing for better reliability of diagnosis?

-can the authors clarify what they mean by risk factor for infection is being infected with pre-Delta … as this wording is confusing (line 37-38)

Major comments:

-can the authors provide the readers a better incite into the public health measures in the local population setting during the different wave time periods at the beginning of the discussion as this helps frame the seroprevalence discussion?

-Despite being an outpatient/non-hospitalised dataset, did the dataset capture those with more severe symptoms that did require hospitalisation?

-is the data generalisable? The authors seem to imply that it is at least generalisable to the US population (line 40-41) but then go on to state that their data seems in opposition to previous reports in the literature for a couple stated reasons, particularly school openings and earlier published data. This was also in opposition to the study stated in Louisiana summer camps, Missouri, Mississippi etc. Why do the authors postulate there are such differences in seroprevalance studies beyond the timeframe of analysis (which I struggle to comprehend why this affects seroprevalence estimates). This goes against the generalisability of the study beyond Texas.

-the discussion doesn’t emphasise what is potentially novel about the presented data, speaks only briefly about vaccinations at the very end (which is arguably one of the more important messages about the manuscript), and doesn’t mention changes in symptomatology or asymptomatic status by variant rather a smorgasbord of prior prevalence studies with variable findings. The authors may consider altering the angle and flow of the discussion

Author Response

Reviewer 2

Comment: population data for those <5 yo would be important particularly given increased prevalence of infection in young children in Omicron waves and important epidemiologically over time. Is this simply a limitation of the data set?

Response: We agree with this comment but yes, participants <5 years old were not included in this dataset. We have added the following statement to the results based on this comment “A fourth limitation of the current analysis is the exclusion of participants less than 5 years old, and thus any result reported here cannot be generalized to that age group.” (lines 94-96).

Comment: How were the time periods of variant determined. Were they local rates of dominant variant >50% in Texas, in the USA, worldwide?

Response:  As stated on lines 148-149, “Variant time periods were verified by Texas Department of State Health Services genetic sequencing data.” Thus, this was Texas-specific and when a variant became >50%.   

Comment: Is a doctor diagnosis reliable especially in subsequent waves with increased community circulation of other viruses (and with such broad definition of symptomatic and less public health measures in Texas emphasising social distancing with open schools and extracurricular activities)? Confirmation with a positive antibody test in acute infection may still be in the window period before antibody development (this seems plausible for those who were actually excluded), if positive antibody this could be past infection with current symptoms representative of a non-SARS-CoV-2 related illness. Were these also confirmed by antigen or PCR testing for better reliability of diagnosis?

Response:  Thank you for these insightful comments. While physician diagnosis without any confirmatory test is not 100% reliable, information is typically collected about household level of infection and if there was another positive case in the household it was assumed this was also a positive case. We did not confirm SARS-CoV-2 cases via antigen or PCR testing in the Texas CARES project but instead relied on self-report data from our participants, which is a limitation of the study.

Comment: Can the authors clarify what they mean by risk factor for infection is being infected with pre-Delta … as this wording is confusing (line 37-38)

Response: Based on this comment, we have rewritten the sentence as follows: “Symptomatic infection increased over time from 47.09% during the pre-Delta variant time period, to 76.95% during Delta, to 84.73% during Omicron, and 94.79% during the Omicron BA.2.”  (lines 34-36)

Comment: Can the authors provide the readers a better incite into the public health measures in the local population setting during the different wave time periods at the beginning of the discussion as this helps frame the seroprevalence discussion?

Response: Based on this comment we have added the following text as a second paragraph to the Discussion: “To give Texas-specific public health mitigation strategy context to the findings re-ported here, on March 19, 2020 Executive Order No. GA-08 was issued by Governor Gregg Abbot, effective March 21 through April 3 with restrictions on certain social activities. Also on March 19, 2020, the Texas Department of State Health Services determined that COVID-19 represents a public health disaster pursuant to the Texas Health and Safety Code. By April 1, 2020 more than 50 Texas counties had issued stay-at-home orders, with many clusters for counties around the most populated Texas cities. This was followed by Gov. Abbott clarifying his previously issued executive order, saying that it “requires all Texans to stay at home” except for essential activities on April 2, 2020. Approximately 1 month later on May 1, 2020 the first phase of the Governor’s plan to allow retail stores, restaurants, malls, and movie theaters to reopen at 25% capacity began followed by sub-sequent phases to completely reopen the state (a detailed timeline can be found at https://www.huschblackwell.com/texas-state-by-state-covid-19-guidance). Schools remained closed for the school year but reopened in August for the 2020-2021 school year with a virtual option available to families. The following two school years saw a subsequent complete relaxation of social distancing and masking requirements.

    The above fairly progressive timeline of public health mitigation strategy reversals may be why our infection results for the pre-Delta variant time period do not generally align with other literature28-36 but are similar to one study in Israel that reported 34% of children with positive status.37 (page 13, lines 42-60)

Comment: Despite being an outpatient/non-hospitalised dataset, did the dataset capture those with more severe symptoms that did require hospitalisation?

Response: Yes, we did ask about symptom severity, and based on this comment have added to Table 1. A summary is provided in the first paragraph of the Results section.  We have also added a Supplementary Table 2 that shows symptom status by variant.

Comment: Is the data generalisable? The authors seem to imply that it is at least generalisable to the US population (line 40-41) but then go on to state that their data seems in opposition to previous reports in the literature for a couple stated reasons, particularly school openings and earlier published data. This was also in opposition to the study stated in Louisiana summer camps, Missouri, Mississippi etc. Why do the authors postulate there are such differences in seroprevalance studies beyond the timeframe of analysis (which I struggle to comprehend why this affects seroprevalence estimates). This goes against the generalisability of the study beyond Texas.

Response: Based on this comment we have added the following sentence to the Limitations section (page 14, lines 104-108): “Also of note, while the study sample represents a large, demographically diverse, and ge-ographically spread state, participants self-selected to participate, which may result in se-lection biases that limit generalizability to the entire population.”

Comment: The discussion doesn’t emphasise what is potentially novel about the presented data, speaks only briefly about vaccinations at the very end (which is arguably one of the more important messages about the manuscript), and doesn’t mention changes in symptomatology or asymptomatic status by variant rather a smorgasbord of prior prevalence studies with variable findings. The authors may consider altering the angle and flow of the discussion.

Response: Based on this comment we have added the following section (and subsequent references) to the Discussion: A second area of key novel findings reported here showed that over four SARS-CoV-2 variant time periods, the proportion of symptomatic infection increased as variants mutated. To the authors’ knowledge, this has not been reported in the literature previously among children not hospitalized with COVID-19 illness. One recently published study using medical claims data in Israel during the third and fourth waves of the pandemic (December 1, 2020, to April 30, 2021, and June 1, 2021, to October 10, 2021, respectively) reported that the rate of children with symptomatic disease among patients with documented SARS-CoV-2 infection was higher in the fourth wave compared to the third wave (49.9% vs. 37.5%).46 They also reported, along with other early-pandemic studies47,48, fever as the most common, and increasing symptom (33% during the fourth wave vs. 13.6% in the third wave) whereas findings here showed fever was more likely to be reported as a component of a constellation of simultaneous symptoms. Specifically, results here showed the most common pre-Delta symptoms or symptom combinations reported were (1) new loss of taste and/or smell, (2) headache, (3) a combination of headache, fatigue, congestion, fever, cough, sore throat, and aches (4) fever, and (5) congestion. This changed during the Delta. Omicron and Omicron BA.2 variant time periods to a combination of congestion, fatigue, headache, fever, cough, sore throat, and aches becoming the most prevalent symptoms reported. Others have reported similar findings in terms of the constellation of symptoms, but earlier in the pandemic. Specifically, one study (December 2020-July 2021) of 109,626 school-aged children in the United Kingdom showed the seven most prevalent symptoms were common to both the Alpha or Pre-Delta and Delta variants.49 The odds of presenting several symptoms were higher with Delta than Alpha infection, including headache and fever. Also of note, the Israeli study mentioned above reported that preschool-aged children had the lowest prevalence of illness compared to other age groups. While our study did not include this age group, our findings were similar in that 10-to-14 year olds and 15-to-19 year olds were significantly more likely than 5-to-9 year olds to be infected with SARS-CoV-2.

New references are as follows:

Ben-Tov A, Lotan R, Gazit S, et al. Dynamics in COVID-19 symptoms during different waves of the pandemic among children infected with SARS-CoV-2 in the ambulatory setting. Eur J Pediatr. 2022;181(9):3309-3318.

Stokes EK, Zambrano LD, Anderson KN, Marder EP, Raz KM, Felix SEB, Tie Y, Fullerton KE. Coronavirus disease 2019 case surveillance—United States, January 22–May 30, 2020. Morb Mortal Wkly Rep. 2020;69:759.

Duarte-Salles T, Vizcaya D, Pistillo A, Casajust P, Sena AG, Lai LYH, Prats-Uribe A, et al. Thirty-day outcomes of children and adolescents with COVID-19: an international experience. Pediatrics. 2021;148:e2020042929.

Molteni E, Sudre CH, Canas LDS, et al. Illness Characteristics of COVID-19 in Children Infected with the SARS-CoV-2 Delta Variant. Children (Basel). 2022;9(5):652.

Round 2

Reviewer 2 Report

The manuscript overall is well written. Although the authors edits improve the manuscript in its present form I still do not find it novel or of particular reader interest nor does it change clinical management (including patient advice), public health measures, or health policy and the results are rather non-generalisable further limiting its utility. The added paragraph to try to address this fails to do so.

Author Response

Comment: The manuscript overall is well written. Although the authors edits improve the manuscript in its present form I still do not find it novel or of particular reader interest nor does it change clinical management (including patient advice), public health measures, or health policy and the results are rather non-generalisable further limiting its utility. The added paragraph to try to address this fails to do so.

Thank you for your kind comment about the manuscript being well written. To address your further concerns, we have added the following paragraph to the end of the Discussion (lines 129-157): “In summary, findings reported here contribute to the current knowledge on pediatric SARS-CoV-2 seroprevalence is several novel ways. First, this analysis is one of the only pediatric longitudinal cohorts in the literature, and in the United States that has included data capture across 4 variant time periods in the pandemic, and that started in 2020 and is still ongoing. While there are other seroprevalence studies published as previously described, few also capture symptom and vaccine status over time and most previous studies occurred early in the pandemic or before the Omicron variant became predominant. This information is important for both clinical management and public health measures, especially as influenza and respiratory syncytial virus (RSV) as well as other common childhood viruses are simultaneously circulating in the population. In other words, it is important for pediatricians and other healthcare providers to have documentation of what symptoms a SARS-CoV-2 infection in a child may be causing now versus earlier in the pandemic to potentially rule out (or in) infection, and how these symptoms may mimic those common to influenza, RSV or other viral infections. This is particularly important information for healthcare providers who have pediatric patients who have never been hospitalized due to COVID-19 illness and thus may not be aware of how SARS-CoV-2 infection symptoms have substantially changed from the beginning of the pandemic. Second, Texas CARES has uniquely captured vaccination uptake data over time that is linked to demographic information. These findings can inform public health vaccination campaigns in terms of populations at high risk for future infection. Our data showed a substantial proportion of children are not protected against future SARS-CoV-2 infection by either naturally acquired antibodies or vaccine-induced so public health efforts should continue to promote vaccination as a preventive measure against future COVID-19 illness.  Third, findings reported here highlight the importance of public health surveillance as an important tool to identify how SARS-CoV-2 seropositivity changes over time in the general population. As we have shown here, the pandemic has evolved in many ways in terms of variants, symptoms produced, and the introduction and uptake of vaccines, and these findings are essential to inform future policy recommendations for not only the current pandemic but others we may be facing in the future.”